# HIV Expression in Infected T Cell Clones

**DOI:** 10.3390/v16010108

**Published:** 2024-01-11

**Authors:** Jason W. Rausch, Shadab Parvez, Sachi Pathak, Adam A. Capoferri, Mary F. Kearney

**Affiliations:** HIV Dynamics and Replication Program, Center for Cancer Research, National Cancer Institute, Frederick, MD 21702, USA; shadab.parvez@nih.gov (S.P.); sachi.pathak@okstate.edu (S.P.); adam.capoferri@nih.gov (A.A.C.); kearneym@mail.nih.gov (M.F.K.)

**Keywords:** human immunodeficiency virus (HIV), persistence, transcriptional regulation, T cell, clonal expansion, HIV rebound

## Abstract

The principal barrier to an HIV-1 cure is the persistence of infected cells harboring replication-competent proviruses despite antiretroviral therapy (ART). HIV-1 transcriptional suppression, referred to as viral latency, is foremost among persistence determinants, as it allows infected cells to evade the cytopathic effects of virion production and killing by cytotoxic T lymphocytes (CTL) and other immune factors. HIV-1 persistence is also governed by cellular proliferation, an innate and essential capacity of CD4+ T cells that both sustains cell populations over time and enables a robust directed response to immunological threats. However, when HIV-1 infects CD4+ T cells, this capacity for proliferation can enable surreptitious HIV-1 propagation without the deleterious effects of viral gene expression in latently infected cells. Over time on ART, the HIV-1 reservoir is shaped by both persistence determinants, with selective forces most often favoring clonally expanded infected cell populations harboring transcriptionally quiescent proviruses. Moreover, if HIV latency is incomplete or sporadically reversed in clonal infected cell populations that are replenished faster than they are depleted, such populations could both persist indefinitely and contribute to low-level persistent viremia during ART and viremic rebound if treatment is withdrawn. In this review, select genetic, epigenetic, cellular, and immunological determinants of viral transcriptional suppression and clonal expansion of HIV-1 reservoir T cells, interdependencies among these determinants, and implications for HIV-1 persistence will be presented and discussed.

## 1. Introduction

Modern ART virtually eliminates the clinical sequelae of HIV-1 infection and typically reduces plasma viremia to levels undetectable by standard clinical assays in people living with HIV (PLWH) [1,2,3]. Yet ART is not curative, as viremia typically returns to levels at or near the initial pre-ART viral set-point within a few weeks of treatment withdrawal, even after years or decades of uninterrupted treatment [4,5]. This viremic rebound originates from infected cells that harbor genetically intact proviruses capable of templating production of new infectious viral particles, referred to collectively as the HIV reservoir [6,7]. This reservoir is mostly comprised of infected CD4+ T cells [8,9]—long-lived master regulators of innate and adaptive immune responses. Infection of cells can be latent, i.e., viral RNA transcription from HIV-1 proviruses integrated into host genomic DNA in these cells is minimal, unproductive, or does not occur [6]. Silencing of viral transcription renders HIV infection minimally disruptive to normal cellular functions and permits evasion of the cytopathicity associated with viral protein production and CTL killing that would otherwise likely result in infected cell death [10,11,12,13,14]. However, the capacity to seed viremic rebound requires either that some cells in the HIV reservoir persist despite incomplete transcriptional suppression, that latency reversal occurs spontaneously and perhaps arbitrarily in some infected cells, or both.

The native and essential capacity of CD4+ T cells for homeostatic or antigen-driven clonal expansion likewise contributes to HIV-1 persistence when manifest in infected cells [15]. Infected T cell proliferation is also sometimes influenced by HIV-1 integration into select human genes [16]. When these cells are latently infected, their proliferation enables expansion of provirus populations while avoiding the deleterious effects of viral gene expression. Moreover, clonal expansion mitigates the risks of sporadic latency reversal, such as may be required for viremic rebound; i.e., if a clonal population of infected cells is replenished or expands more rapidly than it is attritted by spontaneous provirus activation, it will persist indefinitely. Together, HIV-1 transcriptional suppression and infected cell clonal expansion are the two principal determinants of HIV-1 persistence, the primary barrier to a cure for HIV-1 infection. Regulation of and interplay between these two determinants, with emphasis on epigenetic control and in the context of normal CD4+ T cell function, are the topics of this review.

## 2. The HIV-1 Reservoir Is Shaped by Viral Transcriptional Suppression and Clonal Expansion

Administration of ART has profound and lasting effects on HIV-1 population dynamics. Often originating from a single founder virus [17], prolific HIV replication spreads the infection quickly and systemically, thus rapidly expanding the size of the virus population. This unchecked expansion is reflected in extreme viremia during acute infection prior to ART administration, with 10^7^ HIV-1 RNA copies per mL of plasma not uncommon [18,19]. Genetic diversity in the virus population expands rapidly during infection, owing primarily to error-prone reverse transcription [20] coupled with high rates of viral recombination and turnover [21,22,23]. Together, these HIV-1 population features early in infection enable facile selection of adaptive forms that evade selective pressures, potentially including those that result in attenuated viral gene expression. Additionally, prior to ART administration, T cells typically harbor genetically intact proviruses that are likely actively transcribed, generating alternatively spliced or unspliced transcripts that are translated into viral gene products or packaged into nascent virions, respectively. Due to the deleterious consequences of viral protein production, T cells infect pre-ART turnover rapidly, with an estimated half-life of 1–2 days [22,23]. Yet even in the absence of ART, infectious proviruses in some cells are transcriptionally suppressed, and recent evidence indicates that the HIV-1 reservoir population after years or decades on ART is descended from these latently infected cells [24,25].

Administration of effective ART rapidly and systemically halts ongoing virus replication [26,27,28,29,30,31]. However, since neither viral RNA transcription nor translation are among the molecular targets of modern ART, some infected cells capable of producing virions continue to do so even after ART is administered and most die rapidly as a result. These conditions produce marked changes in the dynamics of HIV-1 RNA and DNA populations upon ART initiation. Using the single-copy assay (SCA) [32], viral RNA was shown to decay by approximately 4-logs on ART to an average of 0.6 copies/mL of blood plasma in three phases, attributable to the clearance of infected cells of different types and subsets [33,34,35]. A later study provided evidence for a fourth phase of viral decay on very long-term ART, although the slope appeared to be attributable to only a subset of individuals in the study [36].

More recently, decay rates of defective and predicted-intact viral DNA in peripheral blood mononuclear cells (PBMCs) were dissected in parallel using the Intact Proviral DNA Assay (IPDA [37]) and related approaches [38]. Peripheral blood HIV-1 DNA levels, a reflection of the number of infected cells, also decreased on ART, but more slowly, and with kinetics dependent upon whether proviruses harbored by infected CD4+ T cells were intact or defective. In the specified recent study [38], decay of defective proviral populations had estimated half-lives on the order of several months to a few years, in agreement with prior findings [39,40]. Though not a source of rebound viremia if treatment is withdrawn, persistence of defective proviruses may still lead to CTL recognition and killing [41,42,43], contributing to a state of chronic immune activation often observed in otherwise well-controlled infections [44,45]. In contrast, decay of predicted-intact proviruses on ART was found to be biphasic, with average measured half-lives of 12.9 days and 19.0 months, respectively [38]. Seeming correlation between the second phase of plasma RNA decay and the first phase of viral DNA decay suggests that both may be the result of virus-producing infected T cells in the peripheral blood dying without replacement. Others report that the reservoir decays even more slowly (t_1/2_ = 44 months [46,47,48]) or not at all [49], as forces leading to loss of viral DNA may be almost entirely countered by infected T cell proliferation, increasing their apparent half-life and thereby reducing or preventing clinically meaningful reservoir decay [49].

Taken together, these findings provide important context for the composition of the HIV-1 reservoir upon ART initiation. It is now widely accepted that, in the absence of ongoing rounds of viral replication, HIV-1 population changes are limited almost entirely to dropout of infected cells and skewed by variable rates of cell survival and clonal expansion. Generally, viral DNA populations in PLWH on long-term ART are increasingly dominated by defective proviruses (95–98%) [50,51], and the relatively few persistent intact proviruses are mostly latent [52,53,54] and frequently found in clonally expanded populations [55,56,57]. These viral DNA population characteristics are reflected in plasma viremia typically being reduced to undetectable levels by standard clinical assays (<50 copies HIV RNA/mL). Yet low-level viremia can usually be detected in PLWH on ART using more sensitive laboratory tests [58], and clinically detectable, non-suppressible viremia persists for long periods in some individuals [59]. Moreover, low levels of unspliced intracellular HIV-1 RNA have been detected in variable fractions of clonal infected cell populations across T cell subsets and harboring intact or defective proviruses [60], indicating that at the population level, viral transcriptional suppression is not always complete.

HIV-1 population dynamics in PLWH on long-term ART are ultimately determined by collective viral gene expression, which is itself governed by layers of viral and host genetic, epigenetic, cellular, and immunological determinants, and overarchingly by natural selection; i.e., T cells harboring genetically intact but latent proviruses tend to persist despite ART, while those harboring transcriptionally active intact proviruses typically do not. However, to support viremic rebound upon ART cessation, there must be at least a few cells hidden among latently infected HIV-1 reservoir populations either capable of spontaneous emergence from HIV-1 transcriptional dormancy or within which viral transcription, gene expression, and infectious virion production are not completely silenced—scenarios evidenced by residual low-level viremia and detection of unspliced intracellular RNA [43,51,60]. Escape from this apparent paradox—that some infected cell lineages persist indefinitely despite being capable of producing infectious virions detrimental to their survival—is enabled, in part, by the capacity of HIV-infected cells to clonally expand their populations. More specifically, it is easy to envisage a heterogeneous clonal population of infected T cells wherein, at any given time, most proviruses are latent while a few actively produce virus, the latter group mostly consigned to death by CTL-killing but capable of sparking viremic rebound with cessation of ART. These groups may be distinguishable by epigenetic markers, activation states, or states of T cell differentiation, the determinants of which might be useful therapeutic targets, or perhaps only by the spontaneous emergence of infrequent, stochastically governed intracellular conditions that favor latency reversal. Regardless, identifying the implicated infected cell populations, understanding the bases for their persistence, clonal expansion, and adaptive transcriptional profiles, and how to counter them, are key to curing HIV-1 infection.

## 3. Viral Genetic Determinants of HIV-1 RNA Transcription

A defining feature of the retrovirus lifecycle is reverse transcription of the viral RNA genome into double-stranded DNA, which is then integrated into the host cell genome [61]. HIV-1 favors integration into actively transcribed genes with an accessible chromatin structure, but this preference is far from absolute [62,63]. Consequently, sites of integration in infected CD4+ T cell populations are neither random nor predetermined and are widely distributed. Once strand discontinuities created by the integration process are repaired by host DNA repair machinery, HIV-1 DNA becomes contiguous with and practically indistinguishable from the host cell genome. Accumulated reverse transcription errors and integration site variability are the largest sources of genetic variation in reservoir proviruses with time on ART, with both potentially affecting viral genetic determinants and the cellular genetic and epigenetic contexts of viral gene expression.

HIV-1 genetic elements directly encoded by the provirus and recognized by cellular and viral transcriptional machinery and regulatory proteins collectively constitute the most basic regulatory level of viral gene expression and are summarized schematically in Figure 1. HIV-1 transcription is catalyzed by the cellular enzyme, RNA polymerase II (RNAPII), initiating from the 5′ long terminal repeat (5′LTR) of the provirus, and is dependent upon host cell and viral transcription factors binding to an array of DNA regulatory elements in the 5′LTR promoter [64,65]. The promoter region can be subdivided into four functional domains: (i) a basal core promoter encompassing the transcription initiation site, a CA/TATAA element, and three Sp1 binding sites; (ii) an upstream enhancer region containing two adjacent binding sites for the inducible transcriptional activator nuclear factor kappa B (NF-κB); (iii) an upstream regulatory region involved in cell type-specific expression, and (iv) a downstream regulatory region containing additional secondary enhancer elements and the nidus for assembly of an unstable nucleosome. The LTR promoter also harbors the trans-activation response (TAR) element immediately downstream of the transcription initiation site. Consequent to this positioning, the 5′ termini of all HIV-1 transcripts contain the RNA version of the TAR element [66], which assumes a stable stem-loop structure recognized by the virus-encoded protein Tat via an arginine-rich motif, forming a complex that enables transition from distributive synthesis of short viral RNAs to efficient synthesis of genome-length viral transcripts [67].

The basic mechanism of Tat trans-activation is through Tat-TAR complex recruitment of positive transcription elongation factor b (pTEFb), a general RNAP II elongation factor comprised of cyclin-dependent kinase 9 (CDK9) and Cyclin T1, into proximity of the RNAP II transcription initiation complex [69]. The CDK9 catalytic component of pTEFb mediates phosphorylation of the carboxyl terminal domain of RNAP II, thus activating its capacity for synthesis of long transcripts. This functional transition is further regulated by binding/dissociation of other cellular proteins and protein complexes to the Tat-TAR and RNAP II transcription complexes, CDK9-mediated phosphorylation of these elements, and phosphorylation or acetylation of Tat itself. Relative activity and availability of many of these components in resting versus activated CD4+ T cells, including and especially Tat, CDK9, and Cyclin T1, are important contributors to the propensities toward latency or active viral transcription in the respective T cell states [70,71,72].

Together with TAR, the core promoter elements are sufficient both to promote basal levels of RNAPII-mediated HIV-1 transcription and for Tat-mediated trans-activation. The rate-limiting step in this process is often formation of the transcription preinitiation complex [73,74], i.e., binding of the CA/TATAA element by the general transcription factor TFIID, a large multiprotein complex comprised of the TATA binding protein (TBP) and several associated factors (TAF) [75]. The initiator element downstream of CA/TATAA determines the transcription start site (TSS) and is comprised of two sequence segments between nucleotide positions −6 and +30 [76,77,78,79]. These HIV-1 elements cannot easily be substituted in genetic experiments, suggesting that they have specifically co-evolved to promote viral transcription in cooperation with Tat trans-activation [80,81]. A GC-rich segment immediately upstream of the CA/TATAA element contains three binding sites for Sp1, a ubiquitous zinc finger transcription factor that, in complex with TFIID and Tat, activates processive basal HIV-1 transcription [82,83]. Other host transcription factors (BTEB, Sp3, and Sp4) can bind the Sp1 sites with differing effects [84,85], but only Sp1 coordinates with NF-κB in modulating viral transcription. Lastly, though it resides within the core promoter region, the Oct-1 site antagonizes HIV-1 transcription. Located between the CA/TATAA and initiator elements, Oct-1 transcription factor binding, alone or in conjunction with YY-1, has been shown to repress both basal and Tat-activated viral transcription [86,87,88,89], with a potential role in mediating HIV-1 quiescence in resting T cells [86,90].

The enhancer element is comprised of tandem adjacent NF-κB binding sites immediately upstream of the Sp1 binding sites that modulate basal viral RNA transcription in response to intracellular signaling. Members of the NF-κB family (e.g., NF-κB1, RelA, c-Rel, NF-κB2, and RelB) bind as dimers to this HIV-1 enhancer and select combinations of subunits favor Tat trans-activation [91]. Though not essential for HIV-1 replication, NF-κB binding to the viral enhancer element has been reported to specifically increase the expression of HIV-1 in macrophages and CD4+ T cells [92,93,94,95] activated by proinflammatory cytokines, including TNFα, IL-1β, and IL-6 [92,93,95,96,97,98,99,100,101], anti-inflammatory cytokines such as TGF-β and IL-10 [102], mitogenic stimuli (e.g., phorbol esters [103]), or mycobacterial invasion [104]. Tat itself can also activate NF-κB to further accelerate and sustain virus transcription [105,106,107,108]. The HIV-1 enhancer can bind other transcription factors as well, including Ets-1 [108] and E2F-1 [109], to inhibit or augment the stimulatory activity of NF-κB [110].

Direct binding of transcription factors C/EBP, AP-1, Ets-1, LEF-1, COUP, and NFAT to binding sites in the regulatory region upstream of the enhancer can also up- or down-regulate HIV-1 gene expression in a cell-type-specific fashion [65,111,112,113,114]. Likewise, additional constitutive (Sp1) and inducible transcription factor binding sites (for AP-1, NF-κB, IRF, and NFAT) have been identified in the regulatory region downstream from the transcription start site (TSS), of which several have been shown to be required for efficient HIV-1 replication in T cells [115,116,117,118,119]. AP-1 sites are specifically responsive to T cell activation signals of the cAMP-dependent protein kinase A (PKA) pathway through binding of CREB/ATF proteins [115,117].

Though not comprehensive, this summary illustrates the complexity of viral genetic elements within the 5′LTR and DNA encoding the 5′UTR that contribute to HIV-1 transcriptional regulation even before considering the epigenetic and cellular context. It is not difficult to envision how a provirus containing a few mutations in these regions might be capable of templating synthesis of nascent virions yet sufficiently attenuated to escape self-destruction by cytopathic gene expression or immune killing. Accordingly, as sophisticated, high-throughput methods for sequencing the proviruses comprising the HIV-1 reservoir in donor samples become more widely utilized, close examination of sequence variation in segments of the viral genome capable of directly influencing viral gene expression should be emphasized.

## 4. Epigenetic Determinants of HIV RNA Transcription

Unlike assembly of DNA-bound transcription factors into an RNAP II initiation complex, epigenetic regulation of HIV-1 gene expression is often indirect and involves changing the macromolecular structural environment to favor or disfavor transcription initiation and elongation. Human genomic DNA—including integrated proviruses—is organized into a macromolecular nucleoprotein complex called chromatin, the functional repeating unit of which is the nucleosome. Nucleosomes are comprised of 146 base pairs (bp) of DNA wrapped around an octamer of histone proteins—two each of H2A, H2B, H3, and H4—and are separated by 10–80 bp segments of linker DNA stabilized by histone H1. Strings of linked nucleosomes form chromatin fibers, segments of which can assemble into highly compacted heterochromatin or less compacted euchromatin regions that are generally considered to be restrictive or permissive to gene expression, respectively.

Heterochromatin can be further classified as constitutive or facultative, with each class having important structural and functional distinctions [120]. Constitutive heterochromatin is a highly dense and persistent form of chromatin that exists across cell types and is generally non-permissive toward gene expression. It is comprised primarily of highly repetitive, mostly non-genic sequences localized to centromeres, telomeres, and large segments of chromosomes 1, 9, 16, 19, and Y. Constitutive heterochromatin is thought to be more important for regulating nuclear structure and gene spacing than for storing genetic information and is distinguishable by epigenetic characteristics such as frequent cytosine (CpG) methylation, histone hypoacetylation, and histone H3K9 trimethylation (see below), which provides H3K9me3 sites for heterochromatin protein 1 (HP1) binding. In contrast, though also compact and generally not amenable to RNA transcription, facultative heterochromatin is much more responsive to epigenetic signaling and may be dynamically converted to euchromatin during development, differentiation, or as a natural part of adaptive cellular regulation. Enrichment for trimethylated H3K27 (H3K27me3) is characteristic of facultative heterochromatin.

Although HIV-1 integration favors transcriptionally active regions of the human genome, proviruses have been found integrated into heterochromatic regions, though it remains to be determined how the initial chromatin-free state of the pre-integrative HIV DNA intermediate may affect adjacent chromatin structure, and vice versa. However, as will be described, direct and indirect evidence suggest that both genetic elements within the provirus and the chromatin environment into which it is integrated are important determinants of HIV-1 gene expression and latency. Numerous epigenetic factors contributing to governance of the chromatin environment within and proximal to the integrated HIV-1 provirus have been identified, many of which are discussed below and highlighted in Figure 2.

Nucleosome positioning on the HIV-1 provirus is influenced by DNA sequence, DNA binding proteins, and the activity of chromatin remodelers [121]. From studies on HIV-1 infection in cell culture, it was first determined that three nucleosomes (nuc-0, nuc-1, and nuc-2) assemble at specific positions in transcriptionally inactive proviruses, irrespective of their sites of integration [122,123]. One of these, nuc-1, is deposited immediately downstream of the HIV-1 TSS and thus sterically impedes and is highly repressive of viral transcription. Assembly of nuc-1 is mediated by the BAF chromatin remodeling complex, which is also known as SWI/SWF and is recruited to the LTR by BRD4S despite the DNA sequence in this region of the HIV-1 LTR being predicted to be refractory to nucleosome deposition. Conversely, PBAF—a member of the same family of chromatin remodelers—is required for displacement of nuc-1 and activation of viral transcription. Tat has been reported to play a role in recruitment of PBAF to the 5′LTR [124,125], although PBAF-mediated nuc-1 displacement can also occur through a Tat-independent mechanism [126]. BAF/PBAF involvement in nuc-1 establishment is also linked with integration, as the INI-1 subunit of the BAF complex is known to interact with HIV-1 integrase and perhaps also histone chaperone Spt6 [127,128]. For the same reasons, the NuRD/Mi-2/CHD family of chromatin remodelers has also been linked to HIV latency, as have HIRA (Histone Cell Cycle Regulator) and FACT (Facilitates Chromatin Transcription) complexes [129,130].
Figure 2Epigenetic suppression of HIV-1 RNA transcription. As described in the text, epigenetic host regulatory machinery can contribute to transcriptional suppression of the HIV-1 provirus in numerous ways. Epigenetic suppressive mechanisms generally involve direct or indirect recruitment of histone-modifying enzymes through sequence-specific binding of host proteins or lncRNAs to promoter or regulatory DNA sequences. Though each of the depicted suppressive mechanisms has been demonstrated in in vitro model systems, their relative importance in maintaining latency in infected CD4+ T cells in PLWH remains to be determined. Adapted from [131].
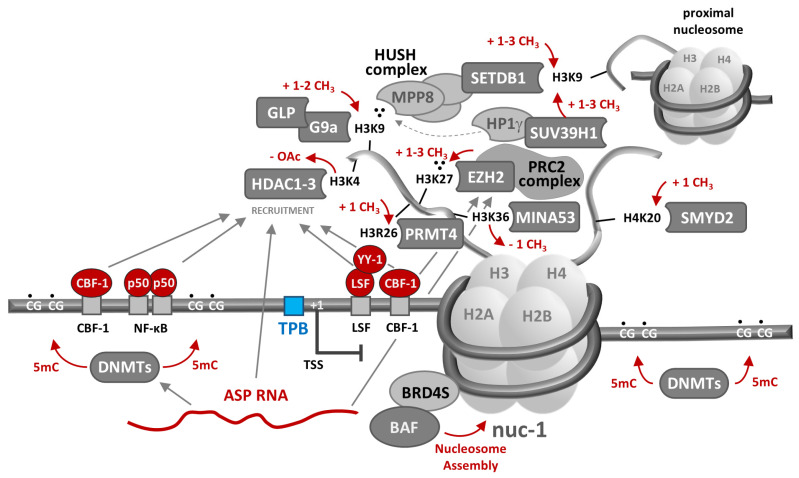


Nucleosome composition and structure, as determined by post-translational modification of constituent histones, is considered a central player in transcriptional regulation in all human cell types as well as in HIV-1 gene expression in infected CD4+ T cells. Histone amino-terminal tails are disordered, protrude from the nucleosome core [132], and are comprised largely of basic amino acids, the sidechains of which may be acetylated, methylated, phosphorylated, ubiquitinated, or sumoylated by different classes of enzymes. These post-translational modifications can affect local chromatin structure, including the degree to which it is compacted, thus regulating promoter accessibility to transcriptional machinery. Modified chromatin can also serve as a scaffold for recruitment of bifunctional effector proteins with bromo-, chromo-, or PHD domains that each can selectively bind specific post-translational modifications [133]. The governing principles that determine how variations in localized collections of post-translational histone modifications affect RNA transcription in different cellular contexts have suitably been referred to as the “histone code” [134].

At least 13 of 70 histone modifiers across 12 human gene families from four classes of enzymes have been specifically implicated in HIV-1 latency [135,136,137,138,139,140,141,142,143]. Histone deacetylases (HDACs) erase the acetylation of lysine ε-amino groups on multiple proteins, including histones, with the effects on the latter target generally considered to be repressive to gene expression [144]. HDAC1, HDAC2, and HDAC3 are class I HDACs shown to be recruited to the HIV-1 5′ LTR during viral latency [89,145,146,147,148] by several transcription factors, including the inhibitory p50-p50 NF-κB homodimer [148], YY-1 and LSF [89,149], and CBF-1 [147]. Once brought into proximity, it has been postulated that these HDACs suppress transcription through deacetylation of lysine 9 of histone 3 (H3K4) in LTR-associated histones. In contrast, SIRT1, a class III sirtuin HDAC, mediates HIV-1 transcriptional control by regulating Tat recycling and transactivation feedback [150].

Histone acetyltransferases (KATs broadly, HATs more specifically) catalyze acetylation of lysine side chains, an activity that, with respect to histones, is generally associated with promoting an open chromatin structure and a favorable transcriptional environment, and thus antagonism of HIV-1 latency. However, contrary to this tendency, lysine acetyl transferase 5 (KAT5/TIP60) has been reported to suppress HIV-1 transcription by acetylating H4K20 on provirus-associated nucleosomes, which in turn recruits bromodomain-containing BRD4 to block elongation of HIV-1 transcription [151].

Possible patterns for histone methylation are numerous and complex, even when limited to those demonstrated to play a role in HIV-1 transcriptional suppression. Two classes of histone methyltransferases (HMT) catalyze mono-, di-, or tri-methylation of basic histone side chains: histone lysine methyl transferases (HKMTs), which can methylate histone tail lysine side chains H3K4, H3K9, H3K27, H3K36, H3K79, and/or H4K20, and protein arginine methyl transferases (PRMTs), which can methylate arginines H3R2, H3R8, H3R17, H3R26, and/or H4R3 [152]. HMT EZH2 is an HKMT, is part of the Polycomb Repressive Complex 2 (PRC2), and catalyzes trimethylation of H3K27, which is associated with heterochromatin and promotes transcriptional suppression [153]. In cell lines and primary cell models of HIV-1 latency, HMT EZH2 and its reaction product (H3K27me3) have been found colocalized to nucleosomes proximal to the viral promoter [139]. Euchromatin HMTs EHMT1/GLP and EHMT2/G9a have likewise been shown to colocalize with their H3K9me2 catalytic products on the HIV-1 promoter in latently infected T cell lines [137,140,141]. These methyltransferases can both be recruited to the 5′LTR by CBF-1, together with HDAC1 and HDAC3 [154], suggesting that viral transcriptional suppression by histone methylation and deacetylation may be coordinated. H3K9 trimethylation can also be spread to or from provirus-proximal nucleosomes by the Human Silencing Hub (HUSH) complex, which is composed of the three subunits TASOR, MPP8, and periphilin [136,142]. H3K9me3 is “read” by the chromodomain-containing MPP8 subunit, which recruits the HKMT “writer” SETDB1 to tri-methylate H3K9 in proximal nucleosomes. Chromodomain-containing HP1g and SUV39H1 bind and spread H3K9me3 across provirus-associated nucleosomes in a similar fashion [138]. SMYD2 mono-methylates H4K20 on the 5′LTR, potentially leading to recruitment of PRC1 and chromatin compaction [135], while CARM1/PRMT4 catalyzes H3R26 arginine methylation to mediate HIV-1 transcriptional suppression—the only PRMT so-implicated to date [143].

Histone demethylases (KDMs) remove methyl groups from histone tail basic residues, countering the activities of HMTs. As with histone acetylation catalyzed by KATs and HATs, histone demethylation is usually associated with an open chromatin structure favoring RNA transcription, although one notable exception has been reported with respect to HIV-1 [155]. MINA53, a KDM, was shown to demethylate H3K36me3 at the HIV-1 5′LTR. This action inhibits binding of KAT8, which specifically recognizes H3K36me3 and catalyzes acetylation of K16 in proximal H4 histones, thereby inducing a local chromatin structural change that favors RNA transcription. Hence, in contradistinction to other enzyme family members, the indirect effect of MINA53 demethylase activity is to promote latency by inhibiting a change in chromatin structure that would favor HIV-1 RNA transcription.

Epigenetic transcriptional suppression may also be achieved by direct methylation of deoxy-cytosines (C→5 mC) in clusters of CpG dinucleotides, called CpG islands (CGI), proximal to cellular promoters [156,157]. There are two such CGIs in HIV-1 promoter and adjacent 5′ non-coding regions of the provirus, respectively, methylation of which has been associated with suppression of viral transcription through recruitment or exclusion of transcription factors and histone modifiers. For example, in a latently infected T cell line model, DNA hypermethylation recruits the methyl-binding protein MBD2, a component of the chromatin remodeling complex NuRD that also contains the histone deacetylase HDAC2 [158]. DNA methylation states of cellular CGIs are established and maintained by DNMT3a/DNMT3b and DNMT1 DNA methyltransferases, respectively [159], though their specific roles in regulating HIV-1 DNA methylation and RNA transcription remain unclear. Further investigation is required to assess the importance of CpG methylation in establishing and maintaining latent infection in PLWH, as data on the topic collected from donor samples are mixed and highly condition-specific [51,158,160,161,162,163]. Nevertheless, the evolutionary retention of CpG islands in transcriptional regulatory regions of the HIV-1 genome suggests that they and their susceptibility to methylation contribute significantly to viral fitness.

Piwi-interacting (piRNAs), long non-coding (lncRNAs), and micro-RNAs (miRNAs) add an additional layer of complexity to HIV-1 transcriptional regulation. Although pi-RNAs were originally thought to be expressed only in the germline where they mediate suppression of transposon activity [164,165], there is now evidence that they, together with the PIWIL proteins involved in their synthesis, are expressed in somatic cells [166]. Together with PIWIL4, piRNAs regulate homeostatic and tumorigenic patterns of gene expression [167] and promote HIV-1 promoter heterochromatinization through the recruitment of SETDB1, HP1, and HDAC4 [168]. Two lncRNAs, NRON and NKILA, indirectly restrict HIV-1 gene expression by inducing proteasomal degradation of Tat [169] or by interfering with the NF-κB signaling pathway [170], respectively. In contrast, MALAT1 and HEAL lncRNAs support HIV-1 transcriptional activity, the former by sequestering the EZH2 subunit and inhibiting the HMT activity of PRC2, and the latter by recruiting HATs to the HIV-1 promoter [171,172,173]. ASP RNA (also called AST) is an HIV-1 antisense transcript that, in cell culture studies, has also been shown to possess lncRNA-like activity, suppressing viral transcription through recruitment of PRC2, EZH2, HDAC1, and DNMT3a, thus indirectly inducing H3K27 trimethylation and nuc-1 assembly [174]. Additional studies to establish the importance of ASP RNA as a mediator of latency in PLWH are underway. Finally, five cellular miRNAs (miR-28, miR-125b, miR-150, miR-223, and miR-382) have been shown to mediate transcriptional suppression of HIV-1 by directly targeting the 3′ ends of HIV-1 mRNAs for degradation [175], and two others (miR-17-5p and miR-20a) by indirectly reducing levels of acetylated Tat [176].

## 5. HIV-1 Transcriptional Suppression and Clonal Expansion in a CD4+ T Cell Functional Context

As regulators of immune responses to diverse and uncertain immunological threats, CD4+ T cells arguably require greater functional plasticity than any other human cell type. This adaptability is manifest in the capacity of these cells to assume any of numerous states of differentiation or effector functionalities in response to immune stimuli, mediated by an exquisitely complex regulatory network of signal transduction. Extracellular environmental signals are transmitted through cell-to-cell contact, cytokine–receptor binding, and antigen recognition complexes to cytoplasmic intermediaries and into the nucleus, where they culminate in transcriptional and epigenetic changes that can persist or be dynamically realigned to fit changing immune conditions. In HIV-1-infected CD4+ T cells in PLWH on ART, the long-term fate of a provirus is inextricably linked to and sometimes dominated by this natural cellular milieu. Accordingly, it is important to consider both viral transcriptional suppression and host cell clonal expansion, the primary determinants of viral persistence, in the context of natural CD4+ T cell phenotypic and regulatory dynamics.

In the thymus early in life and bone marrow later, T cell precursors (thymocytes) differentiate into mature CD4+ T cells expressing unique recombinant T cell receptors (TCRs) on their surfaces. Analogous to antibodies, TCRs recognize specific antigens presented on class II major histocompatibility complexes (MHC II) by antigen-presenting cells (APCs) and must survive positive and negative selection processes to ensure levels of self-antigen recognition sufficient to support homeostatic proliferation yet incapable of precipitating an autoimmune response [177]. Upon maturation, CD4+ T cells are designated naïve because they have yet to encounter sufficient cognate antigen to trigger their activation. Mature naïve CD4+ T cells are released into the circulation, where they migrate via CCR7 homing to secondary lymphoid tissues such as lymph nodes, the tonsils, the spleen, and in various mucous membrane layers in the body, including the gut [178].

Naïve CD4+ T cells are activated when TCR and CD4 co-receptors recognize their cognate antigen in an APC MHC II complex. This activation results in a signaling cascade that drives rapid clonal proliferation and differentiation of activated CD4+ T cells into one of several classes of effector cells depending on the local cytokine milieu, type of APC, concentration of antigen, and co-stimulatory molecules [179,180,181,182]. This process, also called priming to distinguish it from the more rapid secondary activation of memory cells, occurs over 1–2 days and alters the cellular transcription program to endow the T cells with specialized effector functions and a robust proliferative capacity [183]. Polarized CD4+ cell subsets with distinct proliferative capacities and effector functions include T-helper 1 (Th_1_), T-helper 2 (Th_2_), T-helper 17 (Th_17_), follicular helper T cell (T_FH_), induced T-regulatory cells (iT_reg_), and the regulatory type 1 cells (Tr_1_), and potentially distinct classification T-helper 9 (Th_9_) [181].

Once an immune threat has abated and foreign antigen stimulation subsides, activated CD4+ T cells undergo a contraction phase variably reported to last 1–4 weeks [184,185]. This process is characterized by cessation of proliferation and loss of 90–95% of activated cells to apoptotic or non-apoptotic cell death [184,186,187]; however, a small fraction of these cells survive to become long-lived CD4+ memory T cells capable of responding to subsequent immune challenges more rapidly and robustly than during the primary response. Memory cell subsets include stem cell memory (T_SCM_), central memory (T_CM_), transitional memory (T_TM_), effector memory (T_EM_), tissue-resident memory (T_RM_), and recirculating memory (T_RCM_), and are distinguishable by their relative proliferative potential, capacity for specialized effector functions in a secondary response, surface markers, transcriptional profiles, and tissue residence. Resting memory CD4+ T cell populations can be maintained or restored by intermittent, antigen-independent homeostatic proliferation. Homeostatic proliferation requires both IL-7 and IL-15 and is accelerated by interaction with self-antigen presented on APCs, though self-antigen engagement is not essential [188].

CD4+ T cell lineages are inherently plastic and capable of frequent interconversion among memory and effector CD4+ T cell phenotypes, including capacity for activation-driven clonal expansion [189,190,191]. Both phenotypic maintenance and interconversion are driven by antigen and cytokine environments and manifested through engagement of cell surface receptors and downstream signaling cascades, culminating in institution of specialized, phenotype-specific transcriptional programs and chromatin structural changes [192]. There are numerous examples of how effector cell lineage fidelity and memory cell readiness for secondary immune response are durably altered through epigenetic changes [193,194,195,196,197,198,199], and here we relate both a general mechanism and a specific example of how such signal transduction can activate expression of select CD4+ T cell effector genes, either immediately through engagement of proximal enhancers or super-enhancers or more durably by altering chromatin structure (Figure 3) [200]. Notably, CD4+ T cell phenotypes, including capacity for antigen-driven clonal expansion, are regulated by many of the same factors and mechanisms shown to modulate HIV-1 gene expression, as described in previous sections of this review (e.g., SP1, AP-1, BATF, NF-κB, NFAT, and IRF4). This mechanistic overlap, together with chromatin structure around the provirus, can potentially contribute to conflict or synergy between the two primary determinants of HIV persistence in context-specific fashion.

The capacity of CD4+ T cells to transition between permissive and suppressive transcriptional environments serves as the basis for a fundamental model of HIV-1 latency in which cellular and provirus activation states are linked [201]. In accordance with this paradigm, HIV preferentially infects activated CD4+ T cells [202,203], the intracellular environment of which also supports active virus replication that usually results in rapid infected cell turnover. However, when the immune threat that precipitated T cell activation recedes, a few infected cells can survive both contraction and the potentially deleterious effects of HIV-1 infection and persist as infected resting memory cells with transcriptionally suppressive environments that favor viral latency. The relative abundance of cellular transcription factors that enhance viral gene expression (e.g., NF-κB, NFAT, and pTEFb) in activated and resting CD4+ T cells also supports this paradigm [92,204,205,206,207,208,209,210], as does the consistent finding that latently infected cells are most commonly resting memory T cells [8,9,52,53,54], even though these cells are relatively resistant to infection. More precisely, among CD4+ T cell memory subtypes, most infected cells in PLWH on ART are T_SCM_ and T_CM_ [8,211,212], which have the greatest proliferative capacities, while sequence-intact and potentially replication-competent proviruses are more often found in T_EM_ cells [213,214].

Yet the activation states of infected CD4+ memory T cells and the HIV-1 proviruses they harbor are not always correlated [215], and the mechanistic basis for this functional uncoupling remains incompletely understood. Perhaps the most notable example of this disjunction in donor samples is derived from analysis of a very large CD4+ T cell clone harboring a replication-competent provirus [216]. Despite chronic exposure to cognate antigen sufficient to keep the clonal population in a chronic state of activation, only 2.3% of these infected cells were found to express unspliced HIV-1 RNA [60], indicative of uncoupling between cellular and provirus activation states. This notion is further supported by a subsequent study reporting additional instances of extensive antigen-driven expansion of infected cell clones without corresponding high levels of latency reversal [217]. Moreover, in a longitudinal analysis of samples from PLWH on ART, intact HIV-1 proviruses were frequently found in activated CD4+ T cells (i.e., those expressing the HLA-DR activation marker) [218]. One possible explanation for these observations is that some provirus integration sites are not subject to modulation by natural epigenetic regulation of CD4+ T cell activation or differentiation states. This would be consistent with reports of low levels of HIV-1 transcription in infected cells harboring reported-intact proviruses in non-genic regions [219,220].

Like HIV-1 transcriptional suppression, the natural CD4+ memory T cell capacities for homeostatic and antigen-driven proliferation are important enablers of HIV-1 persistence. In homeostatic proliferation, IL-7 expression is upregulated and promotes clonal expansion in uninfected and HIV-1-infected cells alike [221,222]. Moreover, because this mode of infected cell proliferation does not likely induce reactivation of latent proviruses, latently infected clonal T cell populations can expand without being recognized by immune surveillance. However, because the immune function of homeostatic proliferation is to establish and maintain the T cell repertoire, clonal expansion by this mechanism is relatively slow and gradual, particularly in individuals whose HIV infection is well controlled on ART and who do not have other active infections or cancers.

Conversely, antigen-driven clonal expansion of infected CD4+ T cells during activation is a consequence of specific immune challenges, the occurrence and intensity of which are more variable [223,224]. Since cellular proliferation and transition to a more robust transcriptional program are natural components of this activation response, the infected cell transcriptional environment likewise is believed to generally become more favorable for provirus reactivation—an effective reversal of the induced latency model presented above. Pitting the primary determinants of HIV-1 persistence—viral transcriptional silencing and infected cell clonal expansion—against each other in this manner can significantly affect infected cell clonal population dynamics, perhaps best exemplified in studies of HIV-infected CD4+ T cells with HIV-1 antigen specificities (see below). Cells induced to proliferate by immune challenge can become the predominant plasma clones in the infected cell population either intermittently, waxing and waning in concert with escalation and resolution of acute challenges [223,225,226], or remaining elevated when the challenges are chronic [59,216]. Plasma viremia may likewise either transiently spike, producing viremic “blips”, or persist in accordance with this pattern [59].

Clonal expansion dynamics can also vary among infected CD4+ T cells specific for different pathogens. T cells with HIV-1 antigen specificities can become HIV-infected in untreated individuals, though these cells are relatively few and dysfunctional, including impairments in proliferative capacity. Contributors to this dysfunction include provirus activation [227], upregulation of inhibitory molecules [228,229,230], chronic immune activation [231], and loss of lymphoid structure supporting CD4 homeostasis [232,233,234]. These impairments, coupled with the cytopathicity often caused by HIV-1 protein production in these cells [235], can result in clonal depletion, although proliferative responses can be restored by initiation of ART [236]. Proliferation dynamics of CD4+ T cells specific to antigens expressed on other pathogens are likewise distinctive, owing to differing surface receptor and cytokine synthesis and response profiles. Perhaps most notably, and in contrast to HIV-specific CD4+ T cells, cells recognizing cytomegalovirus (CMV) antigens are generally preserved in function, quantity, and proliferative capacity during HIV-1 infection [237,238,239].

## 6. Site of Provirus Integration Can Affect Both Viral Gene Expression and Clonal Expansion in HIV-1-Infected CD4+ T Cells

With improved methodologies [219,220,240,241,242,243], evidence is accumulating that sites of provirus integration can shape HIV-1 reservoir dynamics by influencing both viral gene expression and the propensity of infected cells to clonally expand. Nearly a century ago, it was first demonstrated that DNA translocations within and among chromosomes can have a critical influence on cellular phenotype, an influence dubbed ‘position effects’ [244]. Much more recently, the mechanistic basis for position effects was linked to translocated genes adopting the local chromatin structural environment and predisposition toward gene expression thereof [245]. It should therefore perhaps not be surprising if gene expression from integrated HIV-1 proviruses is determined to have a similar position dependence [246].

HIV-1 integration favors active genes [63], particularly at loci with open, accessible chromatin structures [247,248] associated with the histone marker H3K36me3. Following integration, long-term persistence of the infected CD4+ T cells requires mutual compatibility between provirus and proximal gene expression, and this congruence is likely at least partly enabled by adaptations in local chromatin structure. Human chromosomes can be structurally subdivided into topologically associated domains (TADs) flanked by CCCTC-binding factor (CTCF) binding sites that restrict chromatin spread outside domain boundaries. The borders of smaller compartments within these domains are demarcated both by transcriptional activity and 3D interactions, and these environments are preserved by the structural maintenance of chromosome (SMC) complex SMC5/6 [249]. An important component of this maintenance function is resolution of transcriptionally induced topological stress [250,251], such as can occur when HIV-1 provirus transcriptional orientation opposes that of proximal host genes. Though SMC5/6 has not been specifically implicated in HIV-1 transcriptional silencing, this and other epigenetic mechanisms described in this review contribute to the evolution or preservation of provirus chromatin structure over time, perhaps explaining accumulation of silencing histone modifications correlating with provirus quiescence in cell culture [252]. Conversely, though the commonly accepted paradigm is that the chromatin structure of the provirus conforms to the local structural environment into which it has been integrated, it is conceivable that the reverse may also occur, i.e., that provirus chromatin and transcriptional states impose themselves on proximal host genes via SMC5/6, HUSH complexes, or other mechanisms—an intriguing possibility that might confer a survival advantage to select HIV-1 reservoir cells and thus merits further investigation.

A system of analysis based on the proximity ligation (PLA) recently revealed associations among local histone modifications, Tat recruitment, proviral transcriptional activity, and capacity for latent proviral activation by stimulation in individual cells in vitro [240,253]. When HIV-1 integration occurs in stimulated CD4+ T cells, cell and provirus activation states are typically aligned, though some uncoupling has been observed [254,255]. More specifically, short HIV-1 transcripts are frequently detected irrespective of cell activation, consistent with RNAP II stalling 50 bp after initiation [252,256,257], and their presence is correlated with more rapid viremic rebound after cessation of ART [258,259]. Using PLA-based analysis of HIV-1-infected cells induced to activation ex vivo, it was determined that provirus-proximal histone marks K3K4me1 and H3K27ac were often associated with short viral transcripts prior to activation and Tat-mediated latency reversal afterwards [240,252], providing additional chromatin structural context to related observations [246,247,257,260]. These histone marks are characteristic of the nearly one million enhancer and super-enhancer regions of the human genome, shown to be favored targets of HIV-1 integration [247,260] and capable of templating RNAs that—like the short viral transcripts—are neither spliced nor polyadenylated. In contrast, proviruses harboring heterochromatin marks H3K9me3 and H3K27me3 were not easily induced to transcribe viral RNA, in accordance with other studies [138,158,247], while the H3K4me3 histone modification was associated with transcriptionally active proviruses capable of generating full-length viral transcripts in stimulated infected cells. From these collective observations, it has been postulated that, in contrast to histone marks typically associated with heterochromatin, “enhancer” chromatin modifications may prevent compartmentalization of a provirus into transcriptionally repressive heterochromatin and yet limit viral RNA synthesis to short transcripts in reservoir cells, thus avoiding the deleterious effects of HIV-1 gene expression while remaining “primed” for intermittent provirus activation [252,261].

Although persistent infected cells harboring proviruses in an enhancer chromatin structural context is a plausible source of rebound viremia once ART is halted, recent evidence instead suggests a long-term evolutionary trend for HIV-1 reservoir populations in PLWH toward selection of cells harboring proviruses in non-genic, transcriptionally dormant regions of heterochromatin [219,220]. More specifically, analyses of individual proviruses reported to be intact from both PLWH on ART for more than 20 years and elite controllers, i.e., those few individuals who maintain undetectable levels of plasma viremia without ART, revealed that most were integrated into pericentromeric or repetitive satellite DNA regions of transcriptionally inert constitutive heterochromatin and thus predicted to be highly refractory to reactivation. Such proviruses have been referred to as ‘deeply latent’, and it has been postulated that elite control may be a consequence of an HIV-1 reservoir comprised exclusively of cells harboring ‘deeply latent’ proviruses [262].

In these and other studies [59,219,220,262,263,264,265,266], Krüppel-associated box (KRAB) zinc finger (ZNF) genes have also been shown to be highly represented among genic integration sites of persistent demonstrated- or reported-intact proviruses in PLWH on long-term ART. Acquired over the course of evolution to suppress endogenous transposable elements [267], KRAB-ZNF genes encode bifunctional proteins usually comprised of C-terminal arrays of C2H2-type zinc finger motifs that bind select DNA targets and at least one N-terminal KRAB effector domain, the most common variety of which recruits KAP1/TRIM28 and SETDB1 to locally suppress target gene expression through H3K9 trimethylation [268,269]. Most KRAB-ZNF genes (60%) reside in six gene clusters distributed throughout heterochromatin-enriched chromosome 19, and KRAB-ZNF genes have generally been identified as targets for the HUSH complex, which likewise suppresses gene expression through H3K9 trimethylation [270,271,272]. For these reasons, the chromatin structural environment around KRAB-ZNF genes has been considered generally repressive, and HIV-1 proviruses integrated into KRAB-ZNF genes refractory to induction to a degree comparable to those integrated into pericentromeric constitutive heterochromatin.

There is ample evidence, however, that KRAB-ZNF genes are not constitutively dormant, and both the original and evolved functions of KRAB-ZNF genes suggest a potentially unique niche as a safe harbor for latent replication-competent proviruses capable of re-activation. For instance, in a sample donated by a PLWH on ART for more than 21 years, an infected T cell clone harboring a confirmed-intact provirus integrated into ZNF268 was shown to contribute to that individual’s non-suppressible viremia—a clear indication that this integrant is not ‘deeply latent’ [59]. More recently, two T cell clones harboring confirmed-intact proviruses integrated into ZNF470 and ZNF721 (dubbed ZNF470i and ZNF721i, respectively) were identified in samples donated by an elite controller with metastatic lung cancer, with the latter clone supporting low-level viral transcription when infected cells were induced to activation by cognate antigen ex vivo [266]. Moreover, all three clones harboring intact proviruses in KRAB-ZNF genes were very large relative to the respective infected CD4+ T cell populations of the two respective donors, suggesting that provirus inducibility, infected T cell proliferative potential, and sites of provirus integration are potentially interrelated.

In a comparative study of HIV-1 integration sites identified in primary CD4+ T cells infected ex vivo and samples from PLWH on ART, integration into any of seven genes was found to confer a highly statistically significant proliferative and/or survival advantage to infected cell clonal populations [16]. These genes, i.e., STAT5B, BACH2, MKL2, MKL1, IL2RB, MYB, and POU2F1, are all cancer-associated, and proviruses were found integrated into intronic or non-coding regions of these genes upstream of some or all exons and most often in the same orientation as gene transcription, in contradistinction to the pattern generally observed in persistent infected cell populations in PLWH on ART [216,273,274]. Together, these observations suggest that a survival advantage may be conferred through insertion of an exogenous HIV-1 LTR promoter(s) in non-coding regions of select cancer-related genes, resulting in their overexpression or dysregulation. This, in turn, may reduce or increase infected cell responsiveness to apoptotic or proliferative signals, respectively. Reported associations between HIV-1 integration into STAT3 and LCK genes and development of AIDS-related T cell lymphomas represent perhaps the most extreme example of such HIV-1 integration-induced cellular dysregulation [275].

Yet HIV-1 integration into select cancer-related genes is a relatively minor contributor to clonal expansion and persistence of infected CD4+ T cells in PLWH on long-term ART [16]. Moreover, all implicated proviruses in the cited examples were defective, and thus incapable of contributing to viremic rebound upon ART cessation. In contrast, KRAB-ZNF genes not only appear to be specifically overrepresented among genic integration sites in CD4+ T cells harboring intact proviruses in PLWH on long-term ART, but anecdotal evidence suggests that cells harboring such proviruses in KRAB-ZNF genes may be more clonally expanded than is typically observed for other genic targets [264]. The question of whether enrichment and propensity toward clonal expansion of CD4+ T cells harboring intact proviruses in KRAB-ZNF genes is related to dysregulation of target gene function, a distinct chromatin structural environment, or both, is actively being investigated in the field.

Finally, powerful new single-cell multiomic methodologies have enabled profiling of infected cells in multiple dimensions simultaneously. For instance, using an approach called PheP-seq, it was recently demonstrated that in large, clonally expanded populations, infected T cells harboring intact proviruses frequently express both ensemble signatures of surface markers conferring increased resistance to immune-mediated killing (i.e., CD44, CD28, CD127, and the IL-21 receptor) and increased levels of immune checkpoint markers likely to limit viral RNA transcription [243]. Single-cell DOGMA-seq [276] was used to demonstrate specific epigenetic programs driving four distinct cellular states of HIV-1-infected cells, as well as elevated levels of IKZF3 expression in cells harboring both latent and transcriptionally active proviruses [276]. IKZF3 encodes the transcription factor Aiolos, a driver of Bcl-2 expression [277] and NF-κB signaling [278] and shown to promote survival and proliferation of HIV-infected cells [279]. DOGMA-seq was also used to show that an HIV-1 provirus can act as an ectopic enhancer that increases chromatin accessibility and recruitment of ETS, RUNT, and ZNF family host transcription factors to regions of chromatin spatially proximal to sites of integration [280]. These and other studies that exploit advanced single-cell technologies will be instrumental in developing a comprehensive yet granular model of HIV persistence.

## 7. Conclusions

Because ART halts ongoing cycles of viral replication, comparing HIV-infected CD4+ T cell populations shortly after ART is initiated to those that persist after decades on ART is highly revelatory of determinants most critical to HIV-1 persistence. Foremost among these is HIV-1 transcriptional latency, governed immediately by availability of transcription factors and engagement with viral genetic elements and more durably by local chromatin structural changes and histone modifications in nucleosomes within and proximal to the proviruses. The proviral transcriptional environment can also be influenced by the activation state of the CD4+ T cell host, though recent advances indicate a selective persistence of reservoir cells with time on ART in which cellular and provirus activation states are uncoupled, a phenomenon that may be related to sites of HIV-1 integration and provirus-proximal chromatin environment. With T cell activation comes antigen-driven clonal expansion, a formidable contributor to HIV-1 persistence, particularly when activation is chronic and provirus quiescence is widely—but incompletely—maintained. Such a condition might be characterized by low basal levels or bursts of stochastic provirus activation in clonal infected cell populations, resulting in low-level systemic virion production and occasional infected cell death, the latter effect counterbalanced by cellular proliferation (Figure 4). These observations collectively profile the dynamics of HIV-1 transcriptional regulation and clonal expansion of infected CD4+ T cells, understanding of which is essential if we are to overcome viral persistence despite ART, the principal barrier to a cure for HIV-1 infection.

## Figures and Tables

**Figure 1 viruses-16-00108-f001:**
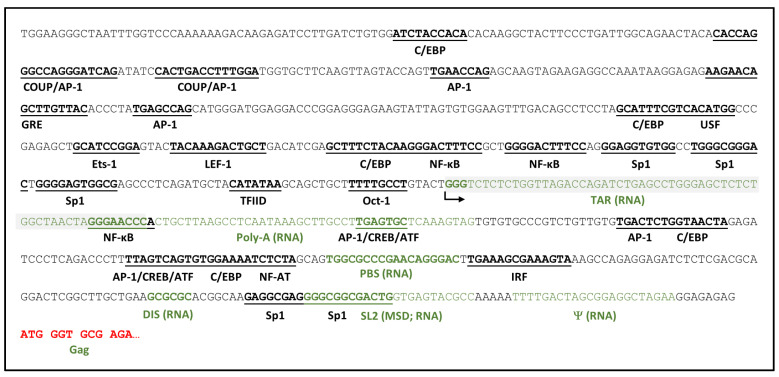
HIV-1 genetic determinants of viral RNA transcription. HIV-1 5′LTR to the initial Gag-coding sequence with transcription factor binding sites and DNA equivalents of select viral RNA elements—TAR, Poly-A signal, PBS, dimerization initiation sequence (DIS), SL2, within which the major splice donor (MSD) is embedded, and packaging signal (Ψ) are highlighted. Arrow denotes the transcription start site. Functions of most elements important for HIV-1 transcriptional control are described in the text. Adapted from [68].

**Figure 3 viruses-16-00108-f003:**
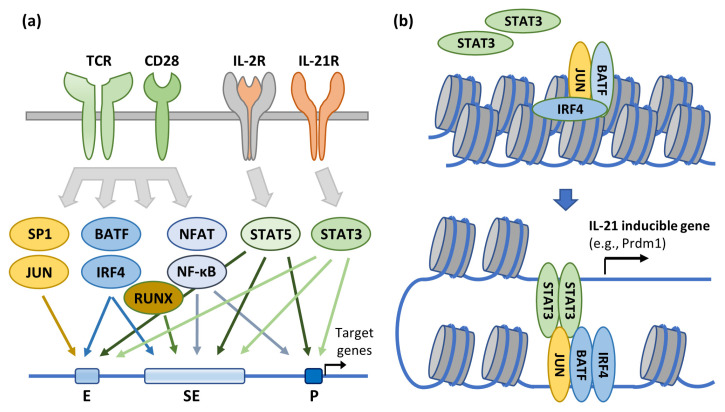
Signal transduction and epigenetic regulation of T cell activation and differentiation. (**a**) Engagement of TCR and co-stimulatory receptor CD28 or cytokine-specific receptors IL-2R and IL-21R activates signaling cascades that determine the genome-wide transcriptional programs that define T cell activation and differentiation states. These effects can be mediated through transcription factor binding to enhancer, super-enhancer, or promoter elements, thereby upregulating target gene expression. (**b**) Select inducible factors can also affect a more durable change in the transcriptional landscape through chromatin remodeling. In this example, the effect is mediated through cooperative binding of AP-1 (a BATF-JUN heterodimer) and IRF-4, which also recruits transcriptional activator STAT3—perhaps via a specific STAT3–JUN interaction. This regulatory mechanism has been specifically demonstrated for upregulation of the IL-21 inducible gene prdm1. Notably, several of these transcription factors (e.g., SP1, AP-1, BATF, NF-κB, NFAT, and IRF4) have also been shown to directly or epigenetically regulate HIV-1 RNA transcription. Adapted from [200].

**Figure 4 viruses-16-00108-f004:**
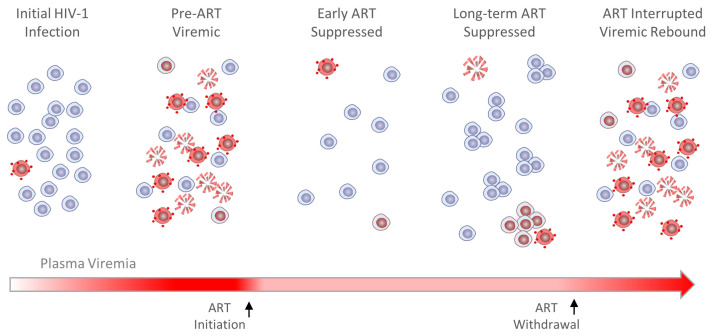
Dynamics of HIV-1 infection and persistence in PLWH on ART. The model depicts the dynamics of uninfected (blue), latently infected (red nucleus, blue cytoplasm), virus-producing (red, with virions), killed (red, fragmented), and clonally expanded (clustered) CD4+ T cells pre-ART, on ART, and after ART is withdrawn. The model considers only infection with replication-competent proviruses. HIV-1 infection typically begins with a burst of virus replication from a single founder virus and spreads rapidly to other CD4+ T cells. Most infected T cells produce virus and are rapidly killed by the cytopathic effects of viral gene expression, a CD8+ T cell immune response, or other immune-killing mechanisms, while the few harboring latent proviruses selectively persist on ART. Likelihood of persistence is further enabled by homeostatic or antigen-driven proliferation, whereby infected T cell populations can expand without the deleterious effects of viral gene expression. Infrequent, sporadic provirus activation in clonally expanded infected CD4+ T cell populations may contribute to rebound viremia within a few weeks after ART is withdrawn.

## Data Availability

Data sharing is not applicable.

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
