# Peer review of "HIV Expression in Infected T Cell Clones"

_viruses, 2024, doi:10.3390/v16010108_

Round 1

Reviewer 1 Report

Comments and Suggestions for Authors

This is a very well written review paper on a subject of high interest. Most of the relevant literature is appropriately cited; the authors may want to double check ref 259 which seems wrong to me. 

Author Response

We thank Reviewer #1 for their comments and commend their attention to detail, as ref 259 was indeed improperly cited. This error has been corrected.

Reviewer 2 Report

Comments and Suggestions for Authors

The Review by Rausch et al for publication at Viruses presents a substantive review of HIV expression in infected T cell clones, providing general overview of our understanding of the latent reservoir as measured in patients, reviewing transcriptional regulatory literature, the epigenetic determinants, and balancing these with our understanding of integration site selection and clonal expansion and T cell development.  The review is welcomed and describes recent paradigms to explain HIV persistence in T cells that is balanced between T cell survival, viral quiescence and clonal proliferation of latently infected cells.  It is richly cited and incorporates both new and old references.  

Of the sections, included the section on viral determinants of transcription is the one section that could use some editing for maximum relevance and updating in light of contemporary understanding of transcription and the relative contributions of both viral and host factors and conserved cis acting elements. 

Figure 1, Viral determinants of transcription, represents a summary of classical transcription studies that defined binding sites in the HIV LTR that can have some effect.  In hindsight, it may be that very few of these have been tested with regard to their importance during productive infection (except perhaps NFkB or NFAT) for their influence on the ability of a provirus to assume a latent or active transcriptional state or its inducibility by different physiological stimuli.  This figure adapted from a 1999 review feels dated and raises questions as to what aspects of this 1999 map are most relevant to the discussion.  

When introducing the viral transcription factor binding sites, we suggest that the authors consider emphasizing factors that are inducible during T cell activation and regulated during T cell development, which are discussed later in the review.  This may help to emphasize the connections to T cell clonal amplification and persistence as a running theme.  

Other sections on Epigenetics, transcriptional suppression, and T-cell lineages are well-written.  

Section 6 is well-written and reviews extensive literature on sites of integration and clonal expansion.  A figure to summarize this section would be helpful as it reviews concepts which are newer and a visual summary of these mechanisms would be welcomed—though not essential.  

Author Response

We thank Reviewer #2 for their thoughtful and constructive suggestions, and concede that much of the information presented in our “Viral genetic determinants of HIV-1 RNA transcription” section is from the relatively early days of research on the topic (not that this necessarily diminishes its importance). One of the challenges in planning and organizing this Review was to define – and limit – its scope, since arguably the subjects of 5 of our 7 sections (i.e., all except “Introduction” and “Conclusions”) could each be covered in sufficient depth to warrant its own dedicated Review. Limiting the scope of the Review required our making choices regarding which topics to cover, in what depth, and the degree to which they are emphasized. One such choice was to briefly introduce non-epigenetic viral determinants of HIV-1 RNA transcription, i.e., Tat/TAR and 5’LTR binding sites for cellular factors, while avoiding more detailed coverage of mechanism and cellular/immunological context and to the relative exclusion of how the cellular factors themselves are regulated – both rich and complex topics also worthy of their own Reviews. We made this choice in part because determining the effects of genetic variation within the HIV-1 LTR on viral populations in infected individuals is a topic of interest in our laboratory that we feel merits more scientific attention than it has received, since, as the Reviewer notes, we still lack full understanding of “what aspects of this 1999 map (of the HIV-1 promoter region) are most relevant to the discussion.” Also, please observe that we do include some discussion of the special importance of NFkB and NFAT for HIV RNA transcription in both this section and subsequently.

We also thank the Reviewer for their suggestion to add a Figure that illustrates how sites of HIV-1 integration can influence clonal expansion. However, Figures describing the correlation between integration into 7 cancer-related genes and clonal expansion have already been published in at least two recent Reviews (PMID 31910871, PMID 34696507), and association between intact provirus integration into KRAB-ZNF genes and clonal expansion remains controversial and has yet to be definitively established. Accordingly, we in this instance chose to graphically depict/describe/summarize other topics we felt would be of greater utility and interest to the reader.

Reviewer 3 Report

Comments and Suggestions for Authors

The presented manuscript thoroughly reviews the mechanisms of HIV expression regulation in infected CD4+ T cells of people living with HIV. The persisting reservoir of cells with the suppressed provirus impedes HIV cure. The authors discuss the notion that clonal proliferation of infected cells, specifically antigen-driven clonal proliferation, has an important role in HIV persistence. This original concept opens up new perspectives on the potential HIV treatment. I would suggest highlighting this idea more and, most importantly, emphasizing that proviral activation is not invariably correlated with antigen-induced cellular activation. In my opinion, the data available on this subject are not stressed enough. To do this, it would be helpful to reorganize Chapter 5, "HIV-1 transcriptional suppression and clonal expansion in a CD4+ T cell functional context." Furthermore, Simonetti, et al., 2020 (doi: 10.1172/JCI145254) research article supports antigenic-driven clonal selection as a significant factor in HIV persistence and should be included in the review. It would also be interesting to consider the reservoir sequence development on ART, which might have supported the extensive clonal proliferation without provirus activation, as shown, for example, by Horsburgh, et al., 2022 (doi: 10.1093/infdis/jiab291). I propose these adjustments for the manuscript to be accepted.

Author Response

We thank Reviewer #3 for their thoughtful and constructive critique. Per their suggestions, we have included the previously overlooked Simonetti, et al. and Horsburgh, et al. references – with brief explanations/discussion – on page 12. We believe this addendum also addresses the Reviewer’s suggestion to increase emphasis on observations of incomplete correlation between cellular and provirus states of activation, that, in lieu of a Chapter/Section 5 reorganization, we hope and trust is a reasonable compromise with respect the degree to which select topics in this Review are discussed and emphasized.